# Valorization of Prickly Pear Peel Residues (*Opuntia ficus-indica*) Using Solid-State Fermentation

**DOI:** 10.3390/foods12234213

**Published:** 2023-11-22

**Authors:** Arturo Coronado-Contreras, Xochitl Ruelas-Chacón, Yadira K. Reyes-Acosta, Miriam Desiree Dávila-Medina, Juan A. Ascacio-Valdés, Leonardo Sepúlveda

**Affiliations:** 1School of Chemistry, Autonomous University of Coahuila, Saltillo 25280, Coahuila, Mexico; scoronado@uadec.edu.mx (A.C.-C.); ykreyes@uadec.edu.mx (Y.K.R.-A.); desireedavila@uadec.edu.mx (M.D.D.-M.); alberto_ascaciovaldes@uadec.edu.mx (J.A.A.-V.); 2Food Science and Technology Department, Autonomous Agrarian University Antonio Narro, Saltillo 25315, Coahuila, Mexico; xruelas@yahoo.com

**Keywords:** hydrolyzable tannins, condensed tannins, *Aspergillus* sp., Box–Hunter and Hunter design, biological activities

## Abstract

Prickly pear peel (*Opuntia ficus-indica*) residues can be used as a substrate in solid-state fermentation to obtain bioactive compounds. The kinetic growth of some *Aspergillus* strains was evaluated. A Box–Hunter and Hunter design to evaluate the independent factors was used. These factors were temperature (°C), inoculum (spores/g), humidity (%), pH, NaNO_3_ (g/L), MgSO_4_ (g/L), KCl (g/L), and KH_2_PO_4_ (g/L). The response factors were the amount of hydrolyzable and condensed tannins. The antioxidant and antimicrobial activity of fermentation extracts was evaluated. *Aspergillus niger* strains GH1 and HT3 were the best for accumulating tannins. The humidity, inoculum, and temperature affect the release of hydrolyzable and condensed tannins. Treatment 13 (low values for temperature, inoculum, NaNO_3_, MgSO_4_; and high values for humidity, pH, KCl, KH_2_PO_4_) resulted in 32.9 mg/g of condensed tannins being obtained; while treatment 16 (high values for all the factors evaluated) resulted in 3.5 mg/g of hydrolyzable tannins being obtained. In addition, the fermented extracts showed higher antioxidant activity compared to the unfermented extracts. Treatments 13 and 16 showed low inhibition of *E. coli*, *Alternaria* sp., and *Botrytis* spp. The solid-state fermentation process involving prickly pear peel residues favors the accumulation of condensed and hydrolyzable tannins, with antioxidant and antifungal activity.

## 1. Introduction

Plants from the Cactaceae family are xerophytes, mostly distributed in desert areas [1]. This family has a great socioeconomic importance. Its members are used as ornamental plants, food, and fodder [2]. Prickly pear fruit, a Cactaceae native to the American continent, are cultivated in Mexico because of its use in gastronomy and can be found in 29 of the 32 states in Mexico [3], and because of its cultural use, Mexico has the largest cactus cultivation area (50,000–70,000 ha); however, it is also cultivated in other continents [4].

Prickly pear cacti are used in the preparation of various foods. The fruit, which is a berry with a thick shell and full of seeds with a mild and sweet flavor, known as prickly pear fruit, has low acidity and is commonly used to prepare beverages (liquors) and sweets. Prickly pear fruit are divided according to their color (green, red, yellow, and purple), which depends on the species and maturation [5]. Due to their production and easy propagation in arid zones, and its application in areas other than food, studies have been conducted on its composition (85% water, 15% sugars, and less than 1% protein). The chemical composition of prickly pear cactus varies depending on the species, the age of the cladodes, the type of soil where it is grown, and the season of the year [6,7,8].

Prickly pear fruit peel is acidic and contains polysaccharides, sterols, lipids, fat-soluble vitamins, and pigments, such as chlorophylls, betalains, coumarins, and carotenoids [9]. These compounds are secondary metabolites and have been previously determined using different methodologies, demonstrating the presence of polyphenols and compounds of interest, such as acids and tannins [10,11,12,13]. Tannins are phenolic compounds that are distributed in the plant kingdom [14]; moreover, they are by-products of plant metabolism that are synthesized in response to external stimuli (stress) [15,16]. They participate in the response or defense by plants against the attack of microorganisms, such as bacteria and fungi. They also take part in the plant’s survival under drought conditions and are classified as hydrolyzable and condensed tannins [17].

Due to the chemical structure of tannins, they can interact with ROS to protect membranes from lipid peroxidation and DNA damage. Furthermore, tannins may act against microorganisms by inhibiting enzymes/substrates or inactivating membrane-bound proteins [18]. Different methods have been used to obtain these compounds, including solid–liquid extraction. This method prefers the use of water as a solvent for environmental reasons, but even so, NaOH, Na_2_CO_3_, and NaHSO_3_ are used, having drawbacks, such as long extraction times, the need for large amounts of solvents, the use of expensive ionic liquids, which makes solute recovery difficult, so solvents such as ethanol, methanol, acetone, Dimethylaminoethyl chloride hydrochloride, Dimethylammonium dimethylcarbamate, and 1-butyl-3-methylimidazole bromide (DIMCARB) are also used. Nevertheless, these solvents are not environmentally friendly, and long durations and high temperatures are still needed for the extraction [19]. These examples are the reason it is necessary to investigate alternative ways to obtain and extract these compounds. An alternative is SSF. Fermentation has been practiced for centuries to produce different foods, such as sufu, tapai, koji, and kimchi. In the case of SSF, a microorganism is cultivated in a solid organic material, where moisture (in the absence or near absence of free water) and a non-soluble material act as a support and nutrient source for the growth of the microorganism, and it has been considered in the last 20 years as an important and viable form of food processing for the bioconversion of agro-industrial waste [20,21].

This bioprocess promotes the bioavailability of the compounds present in the material used, since the microorganisms used can synthesize enzymes that break the cell wall [22]. The fermentation process performs the conversion of complex organic substances into simpler ones, modifying the product physiochemically, improving its quality and the bioavailability of the nutrients present in the substrate [23]. Among the most used microorganisms in SSF are filamentous fungi, such as *Aspergillus*, *Fusarium*, *Penicillium*, *Rhizopus*, and *Trichoderma* [24,25]; although, the use of yeasts and some species of actinobacteria are also reported [26]. SSF is an advantageous method for filamentous fungi, since it is very similar to their natural habitat, which can lead to higher enzymatic productivity compared to submerged fermentation [27]. SSF has advantages such as the low production of wastewater, it does not produce foam, the substrates are low cost (product waste), low substrate volumes, and low moisture content (thus avoiding contamination), but it also shows disadvantages such as heterogeneous media that prevents adequate mixing, moisture levels that are difficult to control, and variables with little precise control (pH, temperature, and dissolved oxygen) [28]. SSF has different applications, such as biodetoxification of agro-industrial waste, the obtaining of enzymes and unicellular proteins, the production of biofuels and biofertilizers, and the obtaining of organic acids such as gallic acid [21,25]. Therefore, the valorization of agro-industrial waste, such as prickly pear peel, is viable for obtaining compounds of interest for their biological activities through SSF [29,30]. Previously SSF has even been performed on prickly pear cacti to improve its protein content, and has been used as fodder, using yeasts such as *Saccharomyces cerevisiae* [31] and *Kluyveromyces marxianus* [32], so it is feasible to ferment prickly pear peel to obtain bioactive compounds, such as tannins. Therefore, the objective of this research was to evaluate the conditions for the SSF process using a Box–Hunter and Hunter experimental design, involving prickly pear peel residues with a strain of *Aspergillus* sp., for the accumulation of condensed and hydrolyzable tannins with antioxidant and antimicrobial activity.

## 2. Materials and Methods

### 2.1. Sample Conditions (Raw Material)

The prickly pear fruit peel was obtained from a local stand (sale of prickly pear cacti), with the raw material coming from Zacatecas, Mexico. Only the prickly pear peel was obtained as residue from this local stand. The peel was washed with a commercial disinfectant solution of sodium hypochlorite, before subsequent freezing at −19 °C until use. After storage, the material was reduced in size by cutting, and dehydrated following the method by Ali et al., 2022 [33], with modifications (oven at 65 °C for 72 h). The dried sample was pulverized in a blender to obtain a fine powder (particle size ≤ 1 mm). After processing, the fine powder was stored in an airtight plastic container at room temperature and kept in a place protected from light.

### 2.2. Physicochemical Analysis of the Raw Material

The critical moisture point and water absorption index were determined following the method by Cerda-Cejudo et al., 2022 [29]. For the critical moisture point, a thermobalance OHAUS^®^ (Nanikon, Switzerland) model MB23 was used, and the total ash was also determined. For the determination of the total sugars, the method by Kejla et al., 2023 [34], was used with some modifications. Specifically, 500 mg of the sample was homogenized with 10 mL of distilled water for 12 h, then 250 µL was placed in a test tube and 250 µL of 5% phenol was added, and subsequently refrigerated for 10 min. Next, 1 mL of concentrated H_2_SO_4_ was added. The mixture was shaken gently in a vortex and placed in a boiling water bath for 5 min, before cooling. Finally, the solution’s absorbance was read by a Thermo Fisher Scientific^TM^ (model: Multiskan^TM^ FC, Waltham, MA, USA) microplate photometer at 480 nm. To determine the total reducing sugars, the DNS method described previously by Prasertsung et al., 2017 [35], was followed and as for the determination of the lipids, the method by Gu et al., 2019 [36], was followed. For the determination of the crude fiber, an acid digestion and a basic digestion were performed; while for the determination of the protein, the Lowry method was used, as described by Deepachandi et al. (2020) [37] in their method.

### 2.3. Growth Kinetics of Filamentous Fungi

Growth kinetics were performed with a humidity of 60% at 30 °C in Petri dishes. The prickly pear fruit peel was dried and ground as a substrate. The strains used were *Aspergillus niger* HT3, *Aspergillus niger* GH1, *Aspergillus niger* Aa20, *Aspergillus niger* Aa210, and *Aspergillus oryzae* sp., which belong to the collection by the Food Research Department at the Autonomous University of Coahuila. To measure the growth of the fungi, the radial growth of the mycelium on the raw material was measured with a Vernier. Once the fungi grew enough to touch one of the ends of the Petri dish, the growth was plotted and the µ_máx_ calculation was performed following the methods by Mitchell et al., 2004, and Ruiz et al., 2012 [38,39]; where X is the radial growth (cm), µ is the maximum specific growth rate constant (1/h), and *t* is the time (h), to determine which strain grows faster on the raw material.

### 2.4. Experimental Design Box–Hunter and Hunter (Fermentation Process and Treatments)

The SSF process was conducted in Petri dishes, together with dried and ground prickly pear peels with *Aspergillus niger* GH1. The fermentation conditions were defined according to a two-level Box–Hunter and Hunter experimental design, that is, a total of 16 treatments, which are shown in Table 1, with the independent factors and the levels used. All the treatments were performed in triplicate, establishing as response factors the accumulation of condensed and hydrolyzable tannins, which were extracted using a solution of absolute ethanol (20 mL). The fermentation extract was recovered with a Whatman filter, using a manual pressing system. The results were analyzed with the statistical package STATISTICA ver. 7.0. (StatSoft. Inc., Tulsa, OK, USA).

### 2.5. Condensed and Hydrolyzable Tannin Determination

The content of hydrolyzable and condensed tannins was measured using the Folin–Ciocalteu reagent, in accordance with De León-Medina et al., 2023 [40], with some modifications. For hydrolyzable tannins, 20 μL of the sample was mixed in a microplate with 20 μL of Folin–Ciocalteu reagent (St. Louis, MO, USA, Sigma-Aldrich) and incubated for 5 min. After that time, the reaction was stopped by the addition of 20 μL of Na_2_CO_3_ 0.01 M and 155 μL of distilled water. The absorbance was measured at 790 nm by a Thermo Fisher Scientific^TM^ (model: Multiskan^TM^ FC, USA) microplate photometer. All the treatments were conducted in triplicate and the results were calculated with a standard solution of gallic acid (3,4,5-Trihydroxybenzoic acid) (Sigma-Aldrich, CAS: 149-91-7). For condensed tannins, 125 μL of the sample was placed in a test tube and mixed with 750 μL of HCl–Butanol (1:9 ratio) with a ferric reagent. The test tubes were sealed, mixed vigorously, and placed in a water bath at 100 °C. After this time, the samples were measured by a Thermo Fisher Scientific^TM^ (model: Multiskan^TM^ FC, USA) microplate photometer at 450 nm. All the treatments were conducted in triplicate and the results were calculated with a standard solution of catechin (Sigma-Aldrich, CAS: 225937-10-0). The hydrolyzable and condensed tannin content was expressed in mg/g dry weight of prickly pear fruit peel.

### 2.6. Antioxidant Activity

The antioxidant activity performed on the different extracts was conducted following three different methods. For the first determination of the antioxidant activity, the method by Sawczuk et al., 2022 [41], with slight modifications was used. A methanolic solution of the DPPH• (1,1-diphenyl-1-picrylhydrazyl) radical was prepared at 60 µM, only methanol was used as a blank and the prepared DPPH- solution was used as a control. A TROLOX (6-hydroxy-2,5,7, 8-tetramethylchroman-2-carboxylic acid) calibration curve was performed from 0 to 250 ppm, adding 96% DPPH• solution and 4% TROLOX solution in each well of the microplate. The samples and the curve absorbance were read with a Thermo Fisher Scientific^TM^ (model: Multiskan^TM^ FC, USA) microplate photometer at 492 nm and the results were expressed in TROLOX equivalents. Likewise, the antioxidant activity was quantified using the free radical ABTS (2,2-azino bis-(-3-ethylbenzothiazolin-6-sulfonate) with some modifications. The reading was performed at 750 nm, up to an absorbance of 0.7 ± 0.02. The TROLOX reagent was used for the curve, as in the previously described method [42]. Finally, the FRAP (ferric reducing antioxidant power) assay was performed, as in the method described by Sik et al., 2022 [43], with some modifications. The readings were performed at 595 nm by the Thermo Fisher Scientific^TM^ (model: Multiskan^TM^ FC, USA) microplate photometer, using TROLOX as standard in the calibration curve in concentrations from 15 to 1000 ppm.

### 2.7. Antimicrobial and Antifungal Activity

Biological tests were performed on the extract with the highest tannin concentration, following the method by Wang et al., 2020 [44], with different microbial strains (*E*. *coli*, *Salmonella*). The crude concentrated extract was evaluated against the strains, using ethanol as a control. The antifungal activity was analyzed using the agar diffusion assay, based on the method by Aqueveque et al., 2017 [45]. The results were reported based on the percentage inhibition of the mycelial growth against *Botrytis* sp. and *Alternaria* sp.

### 2.8. HPLC–MS Analysis of Extracts Fermentation

The analyzes of the fermentation extracts were performed using a Varian HPLC, including an autosampler (Varian ProStar 410, Palo Alto, CA, USA), a ternary pump (Varian ProStar 230I), and a PDA detector (Varian ProStar 330), coupled to a liquid chromatography ion trap mass spectrometer (Varian 500-MSIT mass spectrometer, USA) equipped with an electrospray ion source, following the method described by Cerda-Cejudo et al., 2022 [29], using 0.2 (%, *v*/*v*) formic acid and acetonitrile as the mobile phase at different gradients.

## 3. Results and Discussion

### 3.1. Physicochemical Analysis of Prickly Peer Peel Residues

The results from the proximate analysis were as follows: 37% total sugars, 22% reducing sugars, 6% total proteins, 2% total lipids, 22% total ashes, 14% total fibers, 3% moisture, and a water absorption index of 4.87 g-gel/g dry weight. It is important to clarify that information on the physicochemical parameters for prickly pear peel residues is scarce. However, we can cite some similar works on the cactus cladode [8,10,46,47,48]. Due to the high sugar content and other compounds, this substrate can be used as support in the solid-state fermentation process. In addition, some authors mention that the prickly pear can be used as foodstuffs, for medical applications, and cosmetics [49].

### 3.2. Growth Kinetics of Filamentous Fungi

For the different strains, the growth values were obtained by measuring only the growth on our raw material. The lower values obtained were for *A. niger* Aa20 (0.15 cm/h), *A. oryzae* (0.18 cm/h), and *A. niger* Aa210 (0.20 cm/h). While the fastest strains were *A. niger* GH1 (0.21 cm/h) and *A. niger* HT3 (0.22 cm/h). The *Aspergillus* genus has been widely studied and employed in biotechnological processes due to the extracellular enzyme production from using different agricultural and food industry waste (wheat bran, soybean bran, barley bran, rice straw, corn cob, apple pomace, orange peel, among other) as substrates [50]. Considering the highest values, the two strains with the highest growth were selected (*A*. *niger* GH1 and HT3) for a second kinetic measurement, measuring the release of condensed and hydrolyzable tannins. In Figure 1a, the accumulation of hydrolyzable tannins with the two *A. niger* strains is shown. *A. niger* HT3 achieved a maximum value of 1.5 mg/g at 12 h. While *A. niger* GH1 only achieved a value of 1 mg/g at 12 h. Figure 1b shows that the accumulation of condensed tannins was better for the GH1 strain, with a maximum value of 39.7 mg/g at 60 h. Due to these results, *A*. *niger* GH1 was chosen for the experimental design, since the yields were higher for the condensed tannins, and it had one of the highest µmax values. The fermentation time was 54 h, since between 48 and 60 h of the kinetic measurement, the highest values for the condensed tannins were obtained. This growth on the raw material used was achieved because this fungus has been used previously in samples with compounds very similar to condensed tannins [51]. Furthermore, *A. niger* GH1 was reported as a fungal strain capable of degrading complex substrates (pomegranate, cranberries, and *Larrea tridentata* leaves) for the accumulation of tannins [52].

### 3.3. Evaluation of Box–Hunter and Hunter Design and Response Factors

For the experimental design, the condensed and hydrolyzable tannin accumulation was evaluated. All the treatments were analyzed separately for each response factor. Figure 2 shows the condensed tannin accumulation in the experimental design. Treatments 5, 11, and 13 reached a yield of 35.7, 33.4, and 32.9 mg/g, respectively. There was no significant statistical difference between these treatments. We consider treatment 13 as the best for accumulating condensed tannins; this treatment had the lowest levels of some factors that were evaluated. Figure 3 shows the hydrolyzable tannin accumulation in the experimental design. Treatments 15 and 16 reached a yield of 3.44 and 3.54 mg/g, respectively. There was no significant statistical difference between these treatments. We consider treatment 16 as the best for accumulating hydrolyzable tannins; this treatment had the lowest levels for temperature and some salts that were evaluated.

The factors that showed a significant effect on the tannin accumulation in the fermentation process for both cases were temperature, humidity, and inoculum, as shown in Figure 4. Figure 4a shows the estimated effect on the hydrolyzable tannins, for the factors of humidity (%), inoculum (spores/g), and temperature (°C), which showed a positive effect in the fermentation process. Some authors mentioned that the humidity content is essential for fungal metabolism in the solid-state fermentation process [53]. The dotted line indicates the confidence limit with a value of p 0.05; if the evaluated factors do not exceed this line, it means that they do not directly influence the fermentation process. Therefore, pH, KCl, MgSO_4_, KH_2_PO_4_, and NaNO_3_ do not affect the release of condensed tannins in solid-state fermentation. Figure 4b shows the estimated effect on the condensed tannin accumulation for the factors of temperature (°C) and inoculum (spores/g), which showed a negative effect. On the other hand, temperature is a factor that effects microbial growth. High temperatures can lead to the denaturation of enzymes and effects metabolite production [54]. Finally, humidity (%) showed a positive effect on the fermentation process. As explained above, NaNO_3_, KCl, pH, MgSO_4_, and KH_2_PO_4_ did not effect the release of hydrolyzable tannins in the solid-state fermentation. For both response factors, no significant effect was observed in the salts in the Czapek-Dox medium; this may be due to the nutritional value of the raw material, since its use has been reported to increase nutritional value and to is rich in beneficial compounds [55,56].

### 3.4. Antioxidant Activity in the Fermentation Extracts

The results from the analysis of the antioxidant activity performed using three different methods are shown in Table 2. The samples from the unfermented extract showed the lowest values for antioxidant activity, while the activity in the samples from the fermentation extracts increased considerably. For the ABTS activity, treatment T13 and T16 reached values of 85.59% and 90.61%, respectively. For the DPPH activity, treatment T13 and T16 reached values of 51.29% and 52.05%, respectively. For these tests, the treatments were more effective for the fermented sample compared to the unfermented sample. These results are similar to those by Ali et al., 2022 [33], which obtained values higher than 50% using a known concentration of tannins from prickly pear peel using conventional means. There are no research works related to the fermentation process and antioxidant activity, however, we can mention some similar works. On the other hand, the results reported by Brahmi et al., 2022 [12], who performed ultrasound-assisted extraction, obtained 4078 µmolTE/gDW for DPPH, and 80.8 ± 4.1 µmolTE/gDW for ABTS.

The results on the FRAP activity are shown in Table 2. Treatment T13 and T16 reached values of 1990.78 and 3121.00 mgTEq/L, respectively. These results were up to almost three times more efficient than for the unfermented sample. These results are comparable to those reported by Masmoudi et al., 2021 [57], where the authors reported on the Fe-reducing activity variety at 1245 mg/L to 2045 mg/L; this variation is due to the type of prickly pear used and the extraction method.

### 3.5. Antimicrobial and Antifungal Activity in the Fermentation Extracts

Table 3 shows the bacterial activity against *E. coli*. The inhibition halo formed on the soaked filter paper discs. The control (absolute ethanol) was evaluated, to rule out that the inhibition was caused by the agents other than the fermentation extracts. In the tests with the control, no inhibition halo was observed, while for the fermentation extract belonging to treatment 13, a halo of 13.75 mm was observed. Treatment 16 formed an inhibition halo of 18.33 mm. In another research work, Aruwa et al., 2019 [58], reported an inhibition halo using prickly pear cacti extracts against *E. coli* of 12.2 mm, with known concentrations. These results can be attributed to the different compounds present, such as caffeic acid 4-*O*-glucoside, isorhamnetin, isorhamnetin 3-*O*-glucoside, and *p*-coumaric acid 4-*O*-glucoside. Also, a significant amount of the research on the catechin molecule obtained from different sources, mostly residues, showed pathogenic activity against *E. coli* [59].

In treatment 13, an inhibition zone of 13% was obtained for *Alternaria* sp. While for treatment 16, an inhibition zone of 19% was obtained for the growth of this fungus. These results were compared with the control to discard the possibility that the inhibition occurred due to the solvent. The test against *Botrytis* sp. found that for treatment 13 an inhibition zone of 2% and for treatment 16 an inhibition zone of 9% were obtained.

Similar research with prickly pear fruit residues has already been evaluated for their effect against phytopathogenic fungi. For example, Alqurashi et al., 2022 [59], reported that prickly pear fruit seed oil extracts show activity against the growth of *Saccharomyces cerevisiae* sp., but had no effect on *Aspergillus niger* sp. growth. Brahmi et al., 2020 [60], tested oily extracts from *Opuntia ficus-indica* (L.) Mill. against *Aspergillus niger* 939 N, *Aspergillus flavus* NRRL 3251, *Mucor* sp. NRRL 1829, *Aspergillus ochraceus* NRRL 3174, *Aspergillus parasiticus* CB 5, and *Candida albicans* ATCC 10, with unsatisfactory results. Ammar et al., 2012 [61], reported on the activity against *A. niger* sp. using prickly pear fruit extracts, again attributing this activity to the polyphenolic compounds present in the extracts.

### 3.6. HPLC–MS Analysis of the Fermentation Extracts

The results from the HPLC–RF–MS test are shown in Table 4, where it is possible to appreciate the differences in the compounds found in the unfermented sample and the different treatments. In addition to an increase in the number of compounds found in the treatments in comparison with the unfermented sample. The compounds found in the unfermented sample have been reported previously in works on *Opuntia ficus indica*, such is the case for rhamnetin and isorhamnetin, which was reported by El-Hawary et al., 2020, and Mena et al., 2018 [62,63]. In similar works, quercetin was reported in the fruit peel of *Opuntia* sp. [64]. On the other hand, Benayad et al., 2014, Aruwa et al., 2019, and di Bella et al., 2022 [4,65,66], extracted compounds using different methods, such as solvent extraction and temperature assessment from different parts of *Opuntia ficus-indica*. The fermented sample identified some compounds, which have already been reported as catechin in prickly pear cacti [67,68]. In addition to molecules, such as hydroxytyrosol acetate (3,4-DHPEA-AC), pinocembrin, pterostilbene, scopoletin, and sinensetin, which have been previously reported in other plants but not in *Opuntia ficus-indica*. Some research has reported that the molecules have antioxidant, antiviral, anticancer, and antimicrobial activity [69,70,71].

## 4. Conclusions

*Aspergillus niger* GH1 and HT3 were established as the best strains to accumulate condensed and hydrolyzable tannins, respectively. Treatments 13 and 16 were the best to accumulate condensed (32.9 mg/g) and hydrolyzable tannins (3.5 mg/g), respectively. The humidity (%), inoculum (spores/g), and temperature (°C) effect the release of hydrolyzable and condensed tannins. The fermented extracts showed a higher antioxidant activity compared to the unfermented extracts. Treatments 13 and 16 showed low inhibition of *E. coli*, *Alternaria* sp. and *Botrytis* spp. In the fermentation extracts, up to 12 compounds of phenolic origin were identified. The compounds identified highlight isorhamnetin, *p*-coumaric acid 4-*O*-glucoside, pinocembrin, rhamnetin, among others. The use of solid-state fermentation processes facilitates the accumulation of hydrolyzable and condensed tannins with *Aspergillus* sp., which can be applied in different industrial sectors due to the antioxidant and antimicrobial activity.

## Figures and Tables

**Figure 1 foods-12-04213-f001:**
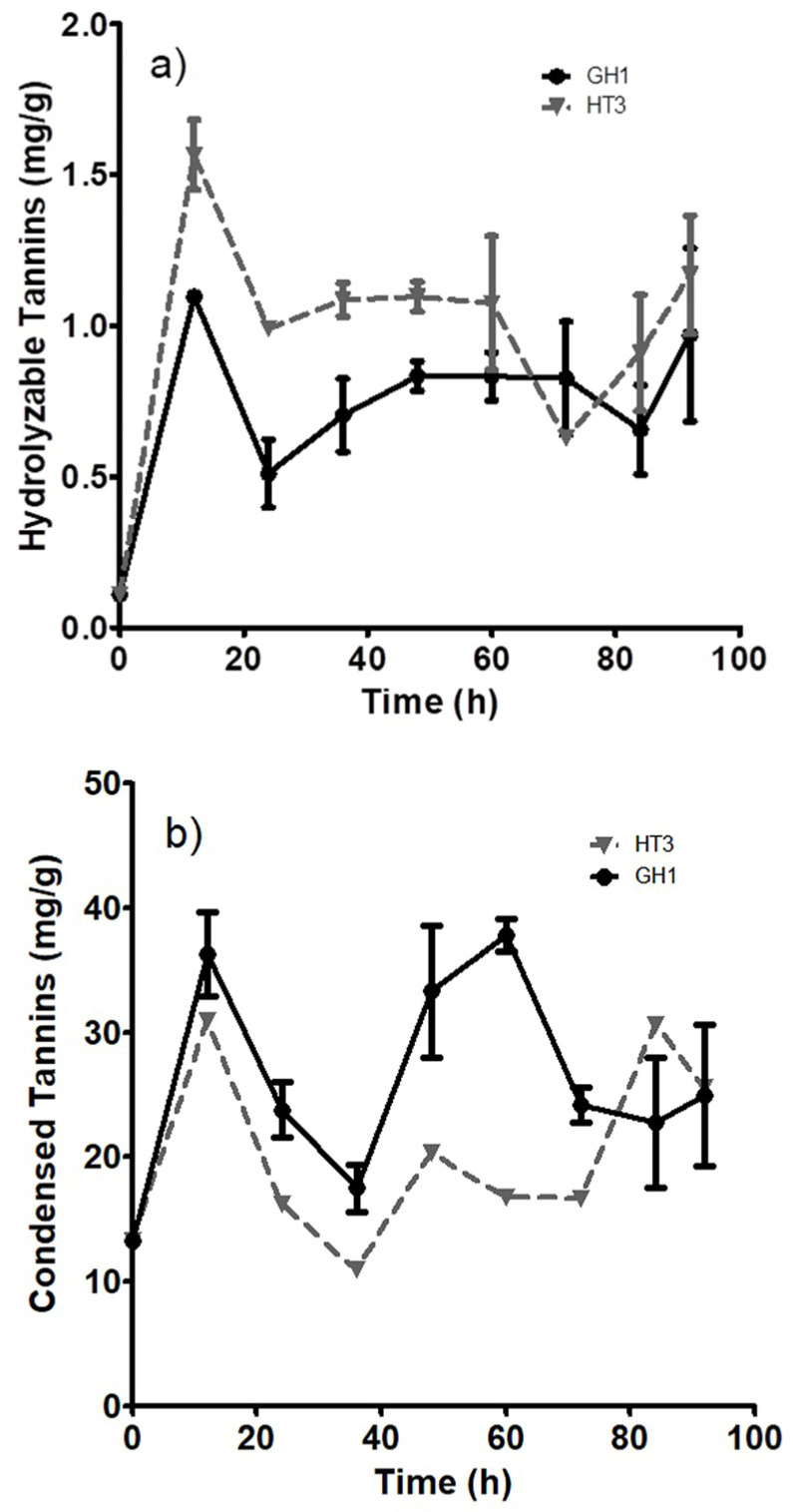
Kinetics of tannin accumulation. (**a**) Hydrolyzable tannins, (**b**) condensed tannins. ● Represents the accumulation performed by *A. niger* GH1, ▼ represents the accumulation by *A. niger* HT3.

**Figure 2 foods-12-04213-f002:**
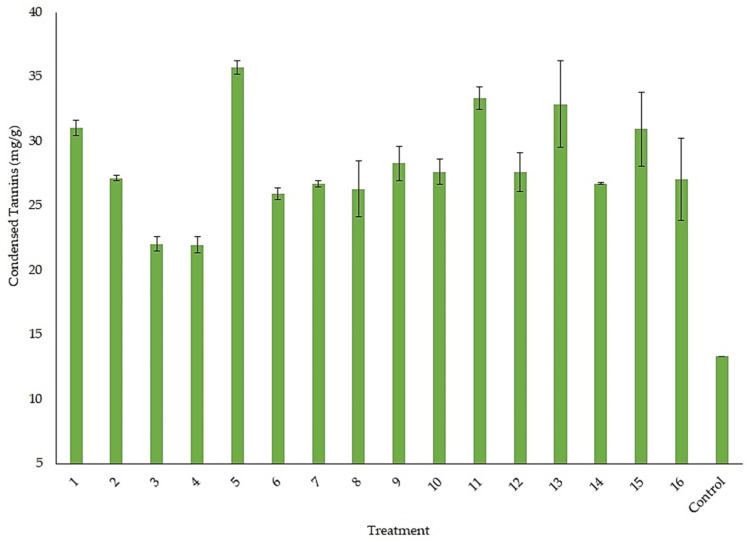
Condensed tannin accumulation for the treatments in the experimental design.

**Figure 3 foods-12-04213-f003:**
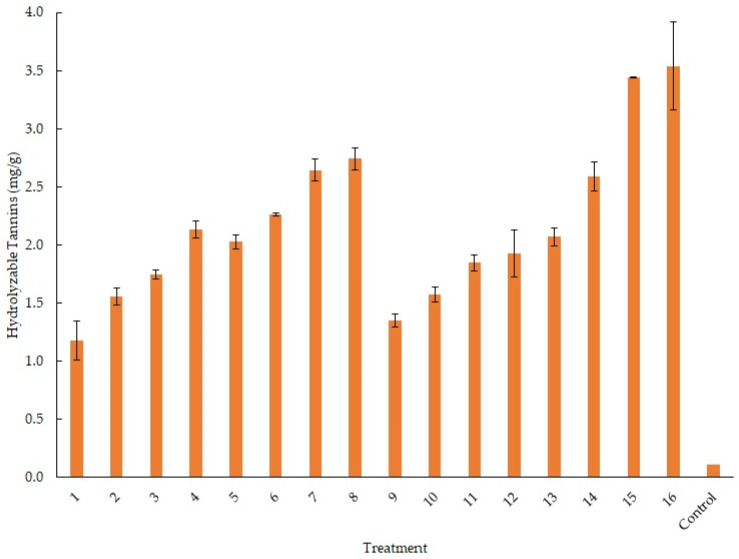
Hydrolyzable tannin accumulation for the treatments in the experimental design.

**Figure 4 foods-12-04213-f004:**
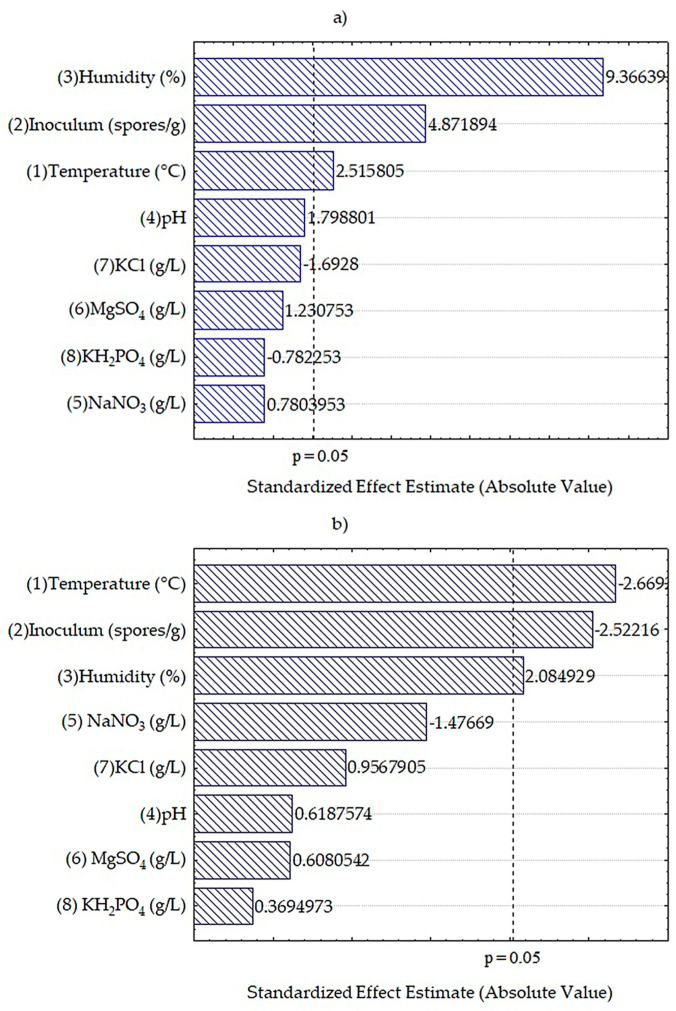
Pareto charts on the factors that influenced tannin accumulation more. (**a**) Hydrolyzable tannins. (**b**) Condensed tannins.

**Table 1 foods-12-04213-t001:** Box–Hunter and Hunter design for the evaluation of the fermentation process conditions, levels, and factors.

Treatment	Temperature (°C)	Inoculum (spores/g)	Humidity (%)	pH	NaNO_3_ (g/L)	MgSO_4_ (g/L)	KCl (g/L)	KH_2_PO_4_ (g/L)
1	−1	−1	−1	−1	−1	−1	−1	−1
2	1	−1	−1	−1	−1	1	1	1
3	−1	1	−1	−1	1	−1	1	1
4	1	1	−1	−1	1	1	−1	−1
5	−1	−1	1	−1	1	1	1	−1
6	1	−1	1	−1	1	−1	−1	1
7	−1	1	1	−1	−1	1	−1	1
8	1	1	1	−1	−1	−1	1	−1
9	−1	−1	−1	1	1	1	−1	1
10	1	−1	−1	1	1	−1	1	−1
11	−1	1	−1	1	−1	1	1	−1
12	1	1	−1	1	−1	−1	−1	1
13	−1	−1	1	1	−1	−1	1	1
14	1	−1	1	1	−1	1	−1	−1
15	−1	1	1	1	1	−1	−1	−1
16	1	1	1	1	1	1	1	1
Factors			Levels					
			+1			−1		
Temperature (°C)			30			25		
Inoculum (spores/g)			1 × 10^7^			1 × 10^6^		
Humidity (%)			70			60		
pH			7			6		
NaNO_3_ (g/L)			6			3		
MgSO_4_ (g/L)			0.52			0.26		
KCl (g/L)			0.52			0.26		
KH_2_PO_4_ (g/L)			1.52			0.52		

**Table 2 foods-12-04213-t002:** Antioxidant activity of the better treatments in the unfermented sample comparison (different letters within each column indicate significant differences at *p* ≤ 0.05. Tukey test).

Samples	ABTS (%)	DPPH (%)	FRAP (mgTEq/L)
Unfermented	25.56 ± 2.14 ^a^	1.77 ± 0.14 ^a^	1070.91 ± 0.94 ^a^
T13	85.59 ± 1.14 ^b^	51.29 ± 1.17 ^b^	1990.78 ± 54.60 ^b^
T16	90.61 ± 0.64 ^c^	52.05 ± 1.55 ^b^	3121.00 ± 58.36 ^c^

**Table 3 foods-12-04213-t003:** Antimicrobial and antifungal activity of the better treatments in the control comparison (different letters within each column indicate significant differences at *p* ≤ 0.05, using the Tukey test).

Samples	*E. coli* (mm Inhibition)	*Botrytis* sp. (% Inhibition)	*Alternaria* sp. (% Inhibition)
T13	13.75 ± 0.28 ^a^	2 ± 0.00 ^a^	13 ± 6.00 ^a^
T16	18.33 ± 0.57 ^b^	9 ± 0.00 ^b^	19 ± 1.00 ^a^

**Table 4 foods-12-04213-t004:** Main compounds identified using HPLC–MS for the best treatments against the unfermented sample. (+) present compound; (−) absent compound.

Retention Time (min)	Metabolite Name	*m*/*z*	Formula	Unfermented Sample	Treatment 13	Treatment 16
5.8	Hydroxytyrosol acetate (3,4-DHPEA-AC)	195	C_10_H_12_O_4_	-	+	+
7.3	Caffeic acid 4-*O*-glucoside	341	C_15_H_18_O_9_	+	-	-
21.3	Pinocembrin	255	C_15_H_12_O_4_	-	+	+
32.1	*p*-Coumaroyl glycolic acid	221	C_11_H_10_O_5_	-	-	+
35.5	Sinensetin	371	C_20_H_20_O_7_	-	+	+
42.8	Isorhamnetin 3-*O*-glucoside	477	C_22_H_22_O_12_	-	+	-
44.1	Apigenin 7-*O*-diglucuronide	621	C_27_H_26_O_17_	-	+	+
46.0	Quercetin	301	C_15_H_10_O_7_	+	-	-
48.6	(+)-Catechin	289	C_15_H_14_O_6_	-	-	+
51.0	Rhamnetin	315	C_16_H_12_O_7_	-	-	+
54.8	Cyanidin	286	C_15_H_11_O_6_	+	-	-
56.1	Isorhamnetin	315	C_16_H_12_O_7_	+	+	+

## Data Availability

The data used to support the findings of this study can be made available by the corresponding author upon request.

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
