# Peer review of "Valorization of Prickly Pear Peel Residues (Opuntia ficus-indica) Using Solid-State Fermentation"

_foods, 2023, doi:10.3390/foods12234213_

Round 1
Reviewer 1 Report
Comments and Suggestions for Authors
This manuscript described the utilization of prickly pear peel residues (Opuntia ficus-indica) through solid-state fermentation to increase tannin content. However, tannins are not the major components in prickly pear. According to the cited reference 10 (J. Agric. Food Chem. 2009, 57, 21, 10323–10330), flavonoid glycosides such as kaempferol and isorhamnetin glycosides were present in this plant but not tannins. Tannins are divided into hydrolysable tannins and condensed tannins. Hydrolysable tannins are sensitive to acid, basic, and fungi fermentation. It was easy to hydrolysis and produce phenolic acid such as gallic acid, ellagic acid, etc. The authors use HPLC-MS and can’t find the above-related compound suggesting it does not contain hydrolysable tannins. In the other way, condensed tannins are easy to condensation from small monomer (catechins) to oligomer by acidic, basic, and oxidation. Color reactions were easy to quantitatively analyze hydrolysable tannins and condensed tannins. But other phenolic compounds such as phenolic acids, and flavonoids also easily interfere with the results. Authors should be more careful to rewrite it otherwise the topic of this manuscript may mislead readers.
Author Response
Dear Reviewer
I appreciate the comments on the manuscript, they help us enrich and improve the document.
Response: We greatly appreciate your comments, they help us improve the manuscript. The document was greatly changed.

Reviewer 2 Report
Comments and Suggestions for Authors
The manuscript entitled “Valorization of prickly pear peel residues (Opuntia ficus-in- 2
dica) using of solid-state fermentation to accumulation of tannins.” Is about the fermentation of prickly pear peel residues. The given information is in this manuscript is useful for researchers and academia and article would be great contribution in related discipline. The room for improvement is always there and I have suggested some minor revisions. Furthermore, the discussion section is very poor and required & related citation to improve the discussion. I can extend my services to further review the incorporation of the corrections in article again.
· Title: reconsider as per your aim and objective - Valorization of prickly pear peel residues (Opuntia ficus-in- 2 dica) using of solid-state fermentation to accumulation of tan- 3 nins
Suggestion-Remove “to accumulation of tan3 nins
· Mention numerical value” These factors were temperature (°C), inoculum (spores/g), humidity (%), pH, NaNO3 14 (g/L), MgSO4 (g/L), KCl (g/L) and KH2PO4 (g/L)
· L-16-17 Need clarity- In addition, the fermentation extracts the antioxidant and antimicrobial 16 activity were evaluated. The results showed that the humidity (%), inoculum (spores/g) and tem- 17 perature (°C) affect the release of hydrolyzable and condensed tannin
· L-105-106 Revise - The Once the sample was 103 dehydrated, it was pulverized in a blender to obtain a more homogeneous raw material. 104 After this processing, the material was stored in an airtight plastic container at room tem- 105 perature and kept in a place protected from ligh
· L-119 Recheck the reference- To determine total reducing sugars, the DNS method described 118 previously by Prasertsung et al., (2017) [35] was followed and as for the determination of 119 lipids, the method of Gu et al., (2019) [36] was followed
· L-172- RECONSIDER Finally, the FRAP (Ferric Reducing Antioxidant 172 Power) assay was performed, as in the method described by Sik et al., (2022) [43] with 173 some modifications, the readings were performed at 595 nm in a microplate reader, using 174 TROLOX as standard in the calibration curve in concentrations from 15 to 1000 ppm, add- 175 ing a ratio of 3% sample and 97% FRAP reagent in each well of the microplate.
· L-191 Revise-headings- 3. Results 191 3.1. Physicochemical analysis of the raw material
· Any scientific logic- . The treatment 13 obtained the highest value for con- 222 densed tannins (43 mg/g). T??????/
· L-239 need clarity- The rest of factors no affect the condensed tannins accumulation in fermen- 239 tation process. F
· Please add more references in discussion section
· Cite the following latest references.
· Carpena, M., Cassani, L., Gomez-Zavaglia, A., Garcia-Perez, P., Seyyedi-Mansour, S., Cao, H., ... & Prieto, M. A. (2023). Application of fermentation for the valorization of residues from Cactaceae family. Food Chemistry, 410, 135369.
· Yafetto, L. (2022). Application of solid-state fermentation by microbial biotechnology for bioprocessing of agro-industrial wastes from 1970 to 2020: A review and bibliometric analysis. Heliyon.
· How can you justify that your study is feasible /economical for the field?
· Please avoid repetition-
· Please check reference style throughout MS
· Italic all the scientific names,
· Remove grammatical mistakes
· Need to rewrite the conclusion
· Recheck Legends description is as per figure number and discussion-
· I urge the authors to improve the English language for better flow of literature
Comments on the Quality of English LanguageModerate editing of English language required
Author Response
Dear Reviewer
I appreciate the comments on the manuscript, they help us enrich and improve the document. Below are the responses to each of your comments.
Title: reconsider as per your aim and objective - Valorization of prickly pear peel residues (Opuntia ficus-indica) using of solid-state fermentation to accumulation of tannins Suggestion-Remove “to accumulation of tannins.
Response: Title was improved.
Mention numerical value” These factors were temperature (°C), inoculum (spores/g), humidity (%), pH, NaNO3 14 (g/L), MgSO4 (g/L), KCl (g/L) and KH2PO4 (g/L).
Response: The abstract was improved.
L-16-17 Need clarity- In addition, the fermentation extracts the antioxidant and antimicrobial 16 activity were evaluated. The results showed that the humidity (%), inoculum (spores/g) and temperature (°C) affect the release of hydrolyzable and condensed tannin.
Response: The abstract was improved.
L-105-106 Revise - The Once the sample was dehydrated, it was pulverized in a blender to obtain a more homogeneous raw material. After this processing, the material was stored in an airtight plastic container at room temperature and kept in a place protected from light.
Response: The phrase was reviewed.
L-119 Recheck the reference- To determine total reducing sugars, the DNS method described previously by Prasertsung et al., (2017) [35] was followed and as for the determination of lipids, the method of Gu et al., (2019) [36] was followed.
Response: The reference was reviewed.
L-172- RECONSIDER Finally, the FRAP (Ferric Reducing Antioxidant Power) assay was performed, as in the method described by Sik et al., (2022) [43] with some modifications, the readings were performed at 595 nm in a microplate reader, using TROLOX as standard in the calibration curve in concentrations from 15 to 1000 ppm, adding a ratio of 3% sample and 97% FRAP reagent in each well of the microplate.
Response: We consider this reference to be important.
L-191 Revise-headings- 3. Results 3.1. Physicochemical analysis of the raw material. Any scientific logic- . The treatment 13 obtained the highest value for condensed tannins (43 mg/g). T??????/
Response: The writing was improved.
L-239 need clarity- The rest of factors no affect the condensed tannins accumulation in fermentation process.
Response: The explanation was improved.
Please add more references in discussion section
Response: Some references were removed and added.
Cite e the following latest references.
Carpena, M., Cassani, L., Gomez-Zavaglia, A., Garcia-Perez, P., Seyyedi-Mansour, S., Cao, H., ... & Prieto, M. A. (2023). Application of fermentation for the valorization of residues from Cactaceae family. Food Chemistry, 410, 135369.
Yafetto, L. (2022). Application of solid-state fermentation by microbial biotechnology for bioprocessing of agro-industrial wastes from 1970 to 2020: A review and bibliometric analysis. Heliyon.
Response: Yaffeto et al was already cited in text (line 77-78). Carpena et al were added.
How can you justify that your study is feasible /economical for the field?
Response: Until now it is difficult to give an answer, it is the first research work carried out in our work group, we have to understand the biotechnological process to produce bioactive molecules and study fermentation on a large scale. For the moment we can establish the bases on the fungal fermentation process to continue advancing in our line of research.
Please avoid repetition-
Response: The writing was improved.
Please check reference style throughout MS
Italic all the scientific names,
Remove grammatical mistakes.
Response: The references was improved.
Need to rewrite the conclusion.
Response: The conclusion was improved.
Recheck Legends description is as per figure number and discussion.
Response: All figures were reviewed.
I urge the authors to improve the English language for better flow of literature.
Response: The writing was revised and improved.

Reviewer 3 Report
Comments and Suggestions for Authors
The manuscript aims to assess the conditions of the SSF process using prickly pear (OFI) peel and a strain of Aspergillus sp. to accumulate tannins that possess antioxidant and antimicrobial properties. The work has some merit but lacks in providing adequate insights and discussion of the results.
Introduction
Consider providing a brief overview of the significance of antioxidant and antimicrobial activities of tannins in the context of current global health or industrial challenges.
While the disadvantages of SSF are mentioned, it might be beneficial also to highlight its advantages in comparison to other methods, especially if it's a central theme of the study.
Materials and Methods
Line 104: Do you have any idea of the particle size?
Line 117: Please provide more details regarding the equipment used (manufacturer, city)
Line 145: Please include more details regarding the software, (company, city)
Line 153: Please include model, manufacturer, city
Line 174: the same issue regarding the equipment
Results and Discussion
Line 191: I believe this is a “Results and Discussion “section and not just a “Results” section.
Line 202: this section (3.2) has no discussion. Please discuss the curves presented in figure 1.
Line 231: No discussion on section 3.3
Line 239: please correct the use of English “of factors no affect”
In the discussion of the results, it might be helpful to delve deeper into the implications of the findings. For instance, what do the specific values obtained in the physicochemical analysis mean for potential applications of the prickly pear peel?
For the growth kinetics section, consider discussing the significance of the specific μmax values obtained for the different strains.
In the section on evaluating the Box Hunter & Hunter design, it might be beneficial to explain the significance of the specific response factors observed briefly.
For the antioxidant activity section, consider discussing the potential implications of the antioxidant activity levels observed. How do these levels compare to other known antioxidants, and what might this mean for potential applications?
Consider providing more context on the significance of the inhibition halos observed in the antimicrobial and antifungal activity section. How do these results compare to other known antimicrobial and antifungal agents?
For the HPLC-MS Analysis section, consider discussing the potential implications of the specific compounds identified. What might these compounds mean for the potential applications of the fermented extracts?
Conclusion
This section is a mere summary of the results with no real conclusion
Consider elaborating slightly on the potential applications in "different industrial sectors." This could provide readers with a clearer understanding of the study's broader implications.
Please re-phrase “"antioxidant activity compared than the unfermented peel extracts"
Table 2: please explain with the “+” and “–“ represent
Figure 4: these Pareto charts are should be in supplementary material
Comments on the Quality of English LanguageA thorough proofreading is recommended to ensure there are no grammatical or syntactical errors.
Consider rephrasing sentences or phrases that might be ambiguous to ensure clarity.
Author Response
Dear Reviewer
I appreciate the comments on the manuscript, they help us enrich and improve the document. Below are the responses to each of your comments.
Introduction
Consider providing a brief overview of the significance of antioxidant and antimicrobial activities of tannins in the context of current global health or industrial challenges.
Response: Reference 18 was rewritten.
While the disadvantages of SSF are mentioned, it might be beneficial also to highlight its advantages in comparison to other methods, especially if it's a central theme of the study.
Response: The advantages are described on lines 80-87.
Materials and Methods
Line 104: Do you have any idea of the particle size?
Response: Added particle size (line 107).
Line 117: Please provide more details regarding the equipment used (manufacturer, city).
Response:
Line 145: Please include more details regarding the software, (company, city).
Response: More details added.
Line 153: Please include model, manufacturer, city.
Response: Added more information.
Line 174: the same issue regarding the equipment
Response: Added more information.
Results and Discussion
Line 191: I believe this is a “Results and Discussion “section and not just a “Results” section.
Response: The subtitle was modified.
Line 202: this section (3.2) has no discussion. Please discuss the curves presented in figure 1.
Response: This section was modified.
Line 231: No discussion on section 3.3.
Response: This section was modified.
Line 239: please correct the use of English “of factors no affect”
Response: This section was modified.
In the discussion of the results, it might be helpful to delve deeper into the implications of the findings. For instance, what do the specific values obtained in the physicochemical analysis mean for potential applications of the prickly pear peel?
Response: Improved discussion in section 3.1.
For the growth kinetics section, consider discussing the significance of the specific μmax values obtained for the different strains.
Response: Improved discussion in section 3.2.
In the section on evaluating the Box Hunter & Hunter design, it might be beneficial to explain the significance of the specific response factors observed briefly.
Response: Improved discussion in section 3.3.
For the antioxidant activity section, consider discussing the potential implications of the antioxidant activity levels observed. How do these levels compare to other known antioxidants, and what might this mean for potential applications?
Response: Improved discussion in section 3.4.
Consider providing more context on the significance of the inhibition halos observed in the antimicrobial and antifungal activity section. How do these results compare to other known antimicrobial and antifungal agents?
Response: Improved discussion in section 3.5.
For the HPLC-MS Analysis section, consider discussing the potential implications of the specific compounds identified. What might these compounds mean for the potential applications of the fermented extracts?
Response: Improved discussion in section 3.6.
Conclusion
This section is a mere summary of the results with no real conclusion.
Consider elaborating slightly on the potential applications in "different industrial sectors." This could provide readers with a clearer understanding of the study's broader implications.
Response: The conclusion was modified.
Please re-phrase “"antioxidant activity compared than the unfermented peel extracts"
Response: The conclusion was modified.
Table 2: please explain with the “+” and “–“ represent.
Response: Moved table 2 to table 4. More information is added.
Figure 4: these Pareto charts are should be in supplementary material.
Response: We consider that we do not have many figures.
